# Gated Recurrent Convolution Neural Network for OCR

**Jianfeng Wang**[*]
Beijing University of Posts and Telecommunications
Beijing 100876, China
jianfengwang1991@gmail.com

**Xiaolin Hu**
Tsinghua National Laboratory for Information Science and Technology (TNList)
Department of Computer Science and Technology
Center for Brain-Inspired Computing Research (CBICR)
Tsinghua University, Beijing 100084, China
xlhu@tsinghua.edu.cn

## Abstract

Optical Character Recognition (OCR) aims to recognize text in natural images. Inspired by a recently proposed model for general image classification, Recurrent Convolution Neural Network (RCNN), we propose a new architecture named Gated RCNN (GRCNN) for solving this problem. Its critical component, Gated Recurrent Convolution Layer (GRCL), is constructed by adding a gate to the Recurrent Convolution Layer (RCL), the critical component of RCNN. The gate controls the context modulation in RCL and balances the feed-forward information and the recurrent information. In addition, an efficient Bidirectional Long Short-Term Memory (BLSTM) is built for sequence modeling. The GRCNN is combined with BLSTM to recognize text in natural images. The entire GRCNN-BLSTM model can be trained end-to-end. Experiments show that the proposed model outperforms existing methods on several benchmark datasets including the IIIT-5K, Street View Text (SVT) and ICDAR.

## 1   Introduction

Reading text in scene images can be regarded as an image sequence recognition task. It is an important problem which has drawn much attention in the past decades. There are two types of scene text recognition tasks: constrained and unconstrained. In constrained text recognition, there is a fixed lexicon or dictionary with known length during inference. In unconstrained text recognition, each word is recognized without a dictionary. Most of the previous works are about the first task.

In recent years, deep neural networks have gained great success in many computer vision tasks [39, 19, 34, 42, 9, 11]. The fast development of deep neural networks inspires researchers to use them to solve the problem of scene text recognition. For example, an end-to-end architecture which combines a convolutional network with a recurrent network is proposed [30]. In this framework, a plain seven-layer-CNN is used as a feature extractor for the input image while a recurrent neural network (RNN) is used for image sequence modeling. For another example, to let the recurrent network focus on the most important segments of incoming features, an end-to-end system which integrates attention mechanism, recurrent network and recursive CNN is developed [21].

---

[*]This work was done when Jianfeng Wang was an intern at Tsinghua University.

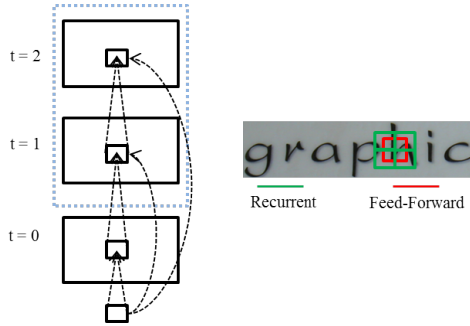

Figure 1: Illustration of using RCL with $T = 2$ for OCR.

A Recurrent Convolution Neural Network (RCNN) is proposed for general image classification [22], which simulates an anatomical fact that recurrent connections are ubiquitously existent in the neocortex. It is believed that the recurrent synapses that exist in the neocortex play an important role in context modulation during visual recognition. A feed-forward model can only capture the context in higher layers where units have larger receptive fields, but this information cannot modulate the units in lower layers which is responsible for recognizing smaller objects. Hence, using recurrent connections within a convolutional layer (called Recurrent Convolution Layer or RCL) can bring context information to all units in this layer. This implements a *nonclassical receptive field* [12, 18] of a biological neuron as the effective receptive field is larger than that determined by feedforward connections. However, with increasing iterations, the size of the effective receptive field will increase unboundedly, which contradicts the biological fact. One needs a mechanism to constrain the growth of the effective receptive field.

In addition, from the viewpoint of performance enhancing, one also needs to control the context modulation of neurons in RCNN. For example, in Figure 1, it is seen that for recognizing a character, not all of the context are useful. When the network recognizes the character "h", the recurrent kernel which covers the other parts of the character is beneficial. However, when the recurrent kernel is enlarged to the parts of other characters, such as "p", the context carried by the kernel is unnecessary. Therefore, we need to weaken the signal that comes from unrelated context and combine the feed-forward information with recurrent information in a flexible way.

To achieve the above goals, we introduce a gate to control the context modulation in each RCL layer, which leads to the Gated Recurrent Convolution Neural Network (GRCNN).

In addition, the recurrent neural network is adopted in the recognition of words in natural images since it is good at sequence modeling. In this work, we choose the Long Short Term Memory (LSTM) as the top layer of the proposed model, which is trained in an end-to-end fashion.

## 2   Related Work

OCR is one of the most important challenges in computer vision and many methods have been proposed for this task. The word locations and scores of detected characters are input into the Pictorial Structure formulation to acquire an optimal configuration of a particular word [26]. A complete OCR system that contains text detection as well as text recognition is designed, and it can be well applied to both unconstrained and constrained text recognition task [3].

The OCR can be understood as a classification task by treating each word in the lexicon as an object category [37]. Another classical method, the Conditional Random Fields, is proposed in text recognition [17, 25, 31]. Besides those conventional methods, some deep network-based methods are also proposed. The Convolutional Neural Network (CNN) is used to extract shared features, which are fed into a character classifier [16]. To further improve the performance, more than one objective functions are used in a CNN-based OCR system [15]. A CNN is combined with the Conditional Random Field graphical model, which can be jointly optimized through back-propagation [14]. Moreover, some works introduced RNN, such as LSTM, to recognize constrained and unconstrained words [30]. The attention mechanism is applied to a stacked RNN on the top of the recursive CNN [21].

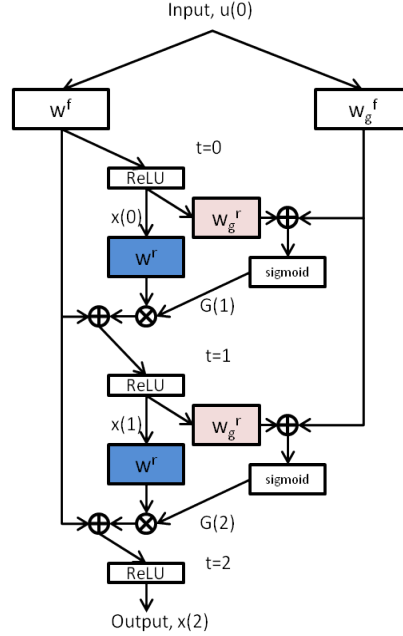

Figure 2: Illustration of GRCL with $T = 2$. The convolutional kernels in the same color use the same weights.

Many new deep neural networks for general image classification have been proposed in these years. The closely related model to the proposed model in this paper is RCNN [22], which is inspired by the observation of abundant recurrent synapses in the brain. It adds recurrent connections within the standard convolutional layers, and the recurrent connections improve the network capacity for object recognition. When unfolded in time, it becomes a CNN with many shortcuts from the bottom layer to upper layers. RCNN has been used to solve other problems such as scene labelling [23], action recognition [35], and speech processing [43]. One related model to RCNN is the recursive neural network [32], in which a recursive layer is unfolded into a stack of layers with tied weights. It can be regarded as an RCNN without shortcuts. Another related model is the DenseNet [11], in which every lower layer is connected to the upper layers. It can be regarded as an RCNN with more shortcuts and without weight sharing. The idea of shortcut have also been explored in the residual network [9].

## 3   GRCNN-BLSTM Model

### 3.1   Recurrent Convolution Layer and RCNN

The RCL [22] is a module with recurrent connections in the convolutional layer of CNN. Consider a generic RNN model with feed-forward input $u(t)$. The internal state $x(t)$ can be defined as:

$$x(t) = \mathcal{F}(u(t), x(t-1), \theta) \tag{1}$$

where the function $\mathcal{F}$ describes the nonlinearity of RNN (e.g. ReLU unit) and $\theta$ is the parameter. The state of RCL evolves over discrete time steps:

$$x(t) = \mathcal{F}((w^f * u(t) + w^r * x(t-1))) \tag{2}$$

where "*" denotes convolution, $u(t)$ and $x(t-1)$ denote the feed-forward input and recurrent input respectively, $w^f$ and $w^r$ denote the feed-forward weights and recurrent weights respectively.

Multiple RCLs together with other types of layers can be stacked into a deep model. A CNN that contains RCL is called RCNN [22].

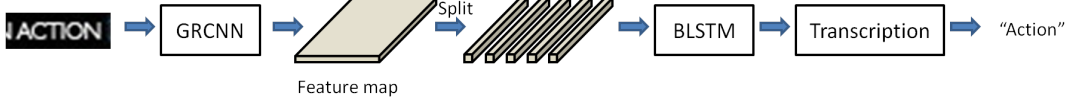

Figure 3: Overall pipeline of the architecture.

## 3.2 Gated Recurrent Convolution Layer and GRCNN

The Gated Recurrent Convolution Layer (GRCL) is the essential module in our framework. This module is equipped with a gate to control the context modulation in RCL and it can weaken or even cut off some irrelevant context information. The gate of GRCL can be written as follows:

$$G(t) = \begin{cases} 0 & t = 0 \\ sigmoid(BN(w_g^f * u(t)) + BN(w_g^r * x(t-1))) & t > 0 \end{cases} \tag{3}$$

Inspired by the Gated Recurrent Unit (GRU) [4], we let the controlling gate receive the signals from the feed-forward input as well as the states at the last time step. We use two $1 \times 1$ kernels, $w_g^f$ and $w_g^r$, to convolve with the feed-forward input and recurrent input separately. $w_g^f$ denotes the feed-forward weights for the gate and $w_g^r$ denotes the recurrent weights for the gate. The recurrent weights are shared over all time steps (Figure 2). Batch normalization (BN) [13] is used to improve the performance and accelerate convergence. The GRCL can be described by:

$$x(t) = \begin{cases} ReLU(BN(w^f * u(t))) & t = 0 \\ ReLU(BN(w^f * u(t)) + BN(BN(w^r * x(t-1)) \odot G(t))) & t > 0 \end{cases} \tag{4}$$

In the equations, "$\odot$" denotes element-wise multiplication. Batch normalization (BN) is applied after each convolution and element-wise multiplication operation. The parameters and statistics in BN are not shared over different time steps. It is assumed that the input to GRCL is the same over time t, which is denoted by $u(0)$. This assumption means that the feed-forward part contributes equally at each time step. It is important to clarify that the time step in GRCL is not identical to the time associated with the sequential data. The time steps denote the iterations in processing the input.

Figure 2 shows the diagram of the GRCL with $T = 2$. When $t = 0$, only the feed-forward computation takes place. At $t = 1$, the gate's output, which is determined by the feed-forward input and the states at the previous time step ($t = 0$), acts on the recurrent component. It can modulate the recurrent signals. Considering two special cases, when all of the output of the gate is 1, it becomes the standard RCL. When all of the output of the gate is 0, the recurrent signal is dropped and it becomes the standard convolutional layer. Therefore, the GRCL is a generalization of RCL and it can adjust context modulation dynamically. The effective receptive field (RF) of each GRCL unit in the previous layer's feature maps expands while the iteration number increases. However, unlike the RCL, some regions that contain unrelated information in large effective RF cannot provide strong signal to the center of RF. This mimics the fact that human eyes only care about the context information that helps the recognition of the objects.

Multiple GRCLs together with other types of layers can be stacked into a deep model. Hereafter, a CNN that contains GRCL is called a GRCNN.

## 3.3 Overall Architecture

The architecture consists of three parts: feature sequence extraction, sequence modelling, and transcription. Figure 3 shows the overall pipeline which can be trained end-to-end.

**Feature Sequence Extraction:** We use the GRCNN in the first part and there are no fully-connected layers. The input to the network is a whole image and the image is resized to fixed length and height.

Table 1: The GRCNN configuration

| Conv | MaxPool | GRCL | MaxPool | GRCL | MaxPool | GRCL | MaxPool | Conv |
|---|---|---|---|---|---|---|---|---|
| $3 \times 3$ | $2 \times 2$ | $3 \times 3$ | $2 \times 2$ | $3 \times 3$ | $2 \times 2$ | $3 \times 3$ | $2 \times 2$ | $2 \times 2$ |
| num: 64 | | num: 64 | | num: 128 | | num: 256 | | num: 512 |
| sh:1 sw:1 | sh:2 sw:2 | sh:1 sw:1 | sh:2 sw:2 | sh:1 sw:1 | sh:2 sw:1 | sh:1 sw:1 | sh:2 sw:1 | sh:1 sw:1 |
| ph:1 pw:1 | ph:0 pw:0 | ph:1 pw:1 | ph:0 pw:0 | ph:1 pw:1 | ph:0 pw:1 | ph:1 pw:1 | ph:0 pw:1 | ph:0 pw:0 |

Specifically, the feature map in the last layer is sliced from left to right by column to form a feature sequence. Therefore, the $i$-th feature vector is formed by concatenating the $i$-th columns of all of the maps. We add some max pooling layers to the network in order to ensure the width of each column is 1. Each feature vector in the sequence represents a rectangle region of the input image, and it can be regarded as the image descriptor for that region. Comparing the GRCL with the basic convolutional layer, we find that each feature vector generated by the GRCL represents a larger region. This feature vector contains more information than the feature vector generated by the basic convolutional layer, and it is beneficial for the recognition of text.

**Sequence Modeling:** An LSTM [10] is used on the top of feature extraction module for sequence modeling. The peephole LSTM is first proposed in [5], whose gates not only receive the inputs from the previous layer, but also from the cell state. We add the peephole LSTM to the top of GRCNN and investigate the effect of peephole connections to the whole network's performance. The inputs to the gates of LSTM can be written as:

$$i = \sigma(W_{xi}x_t + W_{hi}h_{t-1} + \gamma_1 W_{ci}c_{t-1} + b_i), \tag{5}$$

$$f = \sigma(W_{xf}x_t + W_{hf}h_{t-1} + \gamma_2 W_{cf}c_{t-1} + b_f), \tag{6}$$

$$o = \sigma(W_{xo}x_t + W_{ho}h_{t-1} + \gamma_3 W_{co}c_t + b_o), \tag{7}$$

$$\gamma_i \in \{0, 1\}. \tag{8}$$

$\gamma_i$ is defined as an indication factor and its value is 0 or 1. When $\gamma_i$ is equal to 1, the gate receives the modulation of the cell's state.

However, LSTM only considers past events. In fact, the context information from both directions are often complementary. Therefore, we use stacked bidirectional LSTM [29] in our architecture.

**Transcription:** The last part is transcription which converts per-frame predictions to real labels. The Connectionist Temporal Classification (CTC) [8] method is used. Denote the dataset by $S = \{(I, z)\}$, where $I$ is a training image and $z$ is the corresponding ground truth label sequence. The objective function to be minimized is defined as follows:

$$O = - \sum_{(I,z) \in S} \log p(z|I). \tag{9}$$

Given an input image $I$, the prediction of RNN at each time step is denoted by $\pi_t$. The sequence $\pi$ may contain blanks and repeated labels (e.g. $(a - b - -b) = (ab - -bb)$) and we need a concise representation $l$ (the two examples are both reduced to $abb$). Define a function $\beta$ which maps $\pi$ to $l$ by removing blanks and repeated labels. Then

$$p(l|I) = \sum_{\pi:\beta(\pi)=l} \log p(\pi|I), \tag{10}$$

$$p(\pi|I) = \prod_{t=1}^{T} y_{\pi_t}^t, \tag{11}$$

where $y_{\pi_t}^t$ denotes the probability of generating label $\pi_t$ at time step $t$.

After training, for lexicon-free transcription, the predicted label sequence for a test image $I$ is obtained by [30]:

$$l^* = \beta(\arg\max_{\pi} p(\pi|I)). \tag{12}$$

The lexicon-based method needs a dictionary or lexicon. Each test image is associated with a fix length lexicon $D$. The result is obtained by choosing the sequence in the lexicon that has highest conditional probability [30]:

$$l^* = \arg\max_{l \in D} p(l|I). \tag{13}$$

Table 2: Model analysis over the IIT5K and SVT (%). Mean and standard deviation of the results are reported.

(a) GRCNN analysis

| Model | IIIT5K | SVT |
|---|---|---|
| Plain CNN | 77.21±0.54 | 77.69±0.59 |
| RCNN(1 iter) | 77.64±0.58 | 78.23±0.56 |
| RCNN(2 iters) | 78.17±0.56 | 79.11±0.63 |
| RCNN(3 iters) | 78.94±0.61 | 79.76±0.59 |
| GRCNN(1 iter) | 77.92±0.57 | 78.67±0.53 |
| GRCNN(2 iters) | 79.42±0.63 | 79.89±0.64 |
| GRCNN(3 iters) | 80.21±0.57 | 80.98±0.60 |

(b) LSTM's variants analysis

| LSTM variants | IIIT5K | SVT |
|---|---|---|
| $\text{LSTM}_{\{\gamma_1=0,\gamma_2=0,\gamma_3=0\}}$ | 77.92±0.57 | 78.67±0.53 |
| $\text{LSTM-F}_{\{\gamma_1=0,\gamma_2=1,\gamma_3=0\}}$ | 77.26±0.61 | 78.23±0.53 |
| $\text{LSTM-I}_{\{\gamma_1=1,\gamma_2=0,\gamma_3=0\}}$ | 76.84±0.58 | 76.89±0.63 |
| $\text{LSTM-O}_{\{\gamma_1=0,\gamma_2=0,\gamma_3=1\}}$ | 76.91±0.64 | 78.65±0.56 |
| $\text{LSTM-A}_{\{\gamma_1=1,\gamma_2=1,\gamma_3=1\}}$ | 76.52±0.66 | 77.88±0.59 |

# 4 Experiments

## 4.1 Datasets

**ICDAR2003:** ICDAR2003 [24] contains 251 scene images and there are 860 cropped images of the words. We perform unconstrained text recognition and constrained text recognition on this dataset. Each image is associated with a 50-word lexicon defined by wang et al. [36]. The full lexicon is composed of all per-image lexicons.

**IIIT5K:** This dataset has 3000 cropped testing word images and 2000 cropped training images collected from the Internet [31]. Each image has a lexicon of 50 words and a lexicon of 1000 words.

**Street View Text (SVT):** This dataset has 647 cropped word images from Google Street View [36]. We use the 50-word lexicon defined by Wang et al [36] in our experiment.

**Synth90k:** This dataset contains around 7 million training images, 800k validation images and 900k test images [15]. All of the word images are generated by a synthetic text engine and are highly realistic.

When evaluating the performance of our model on those benchmark dataset, we follow the evaluation protocol in [36]. We perform recognition on the words that contain only alphanumeric characters (A-Z and 0-9) and at least three characters. All recognition results are case-insensitive.

## 4.2 Implementation Details

The configuration of the network is listed in Table 1, where "sh" denotes the stride of the kernel along the height; "sw" denotes the stride along the width; "ph" and "pw" denote the padding value of height and width respectively; and "num" denotes the number of feature maps. The input is a gray-scale image which is resized to $100 \times 32$. Before input to the network, the pixel values are rescaled to the range (-1, 1). The final output of the feature extractor is a feature sequence of 26 frames. The recurrent layer is a bidirectional LSTM with 512 units without dropout. The ADADELTA method [41] is used for training with the parameter $\rho$=0.9. The batch size is set to 192 and training is stopped after 300k iterations.

All of the networks and related LSTM variants are trained on the training set of Synth90k. The validation set of Synth90k is used for model selection. When a model is selected in this way, its parameters are fixed and it is directly tested on other datasets (ICDAR2003, IIIT5K and SVT datasets) without finetuning. The code and pre-trained model will be released at `https://github.com/Jianfeng1991/GRCNN-for-OCR`.

## 4.3 Explorative Study

We empirically analyze the performance of the proposed model. The results are listed in Table 2. To ensure robust comparison, for each configuration, after convergence during training, a different model is saved at every 3000 iterations. We select ten models which perform the best on the Synth90k's validation set, and report the mean accuracy as well as the standard deviation on each tested dataset.

Table 3: The text recognition accuracies in natural images. "50","1k" and "Full" denote the lexicon size used for lexicon-based recognition task. The dataset without lexicon size means the unconstrained text recognition

| Method | SVT-50 | SVT | IIIT5K-50 | IIIT5K-1k | IIIT5K | IC03-50 | IC03-Full | IC03 |
|---|---|---|---|---|---|---|---|---|
| ABBYY [36] | 35.0% | - | 24.3% | - | - | 56.0% | 55.0% | - |
| wang et al. [36] | 57.0% | - | - | - | - | 76.0% | 62.0% | - |
| Mishra et al. [25] | 73.2% | - | - | - | - | 81.8% | 67.8% | - |
| Novikova et al. [27] | 72.9% | - | 64.1% | 57.5% | - | 82.8% | - | - |
| wang et al. [38] | 70.0% | - | - | - | - | 90.0% | 84.0% | - |
| Bissacco et al. [3] | 90.4% | 78.0% | - | - | - | - | - | - |
| Goel et al. [6] | 77.3% | - | - | - | - | 89.7% | - | - |
| Alsharif [2] | 74.3% | - | - | - | - | 93.1% | 88.6% | - |
| Almazan et al. [1] | 89.2% | - | 91.2% | 82.1% | - | - | - | - |
| Lee et al. [20] | 80.0% | - | - | - | - | 88.0% | 76.0% | - |
| Yao et al. [40] | 75.9% | - | 80.2% | 69.3% | - | 88.5% | 80.3% | - |
| Rodriguez et al. [28] | 70.0% | - | 76.1% | 57.4% | - | - | - | - |
| Jaderberg et al. [16] | 86.1% | - | - | - | - | 96.2% | 91.5% | - |
| Su and Lu et al. [33] | 83.0% | - | - | - | - | 92.0% | 82.0% | - |
| Gordo [7] | 90.7% | - | 93.3% | 86.6% | - | - | - | - |
| Jaderberg et al. [14] | 93.2% | 71.1% | 95.5% | 89.6% | - | 97.8% | 97.0% | 89.6% |
| Baoguang et al. [30] | **96.4%** | 80.8% | 97.6% | 94.4% | 78.2% | 98.7% | 97.6% | 89.4% |
| Chen-Yu et al. [21] | 96.3% | 80.7% | 96.8% | 94.4% | 78.4% | 97.9% | 97.0% | 88.7% |
| ResNet-BLSTM | 96.0% | 80.2% | 97.5% | 94.9% | 79.2% | 98.1% | 97.3% | 89.9% |
| Ours | 96.3% | **81.5%** | **98.0%** | **95.6%** | **80.8%** | **98.8%** | **97.8%** | **91.2%** |

First, a purely feed-forward CNN is constructed for comparison. To make this CNN have approximately the same number of parameters as GRCNN and RCNN, we use two convolutional layers to replace each GRCL in Table 1, and each of them has the same number of feature maps as the corresponding GRCL. Besides, this plain CNN has the same depth as the GRCNN with $T = 1$. The results show that the plain CNN has lower accuracy than both RCNN and GRCNN.

Second, we compare GRCNN and RCNN to investigate the effect of adding gate to the RCL. RCNN is constructed by replacing GRCL in Table 1 with RCL. Each RCL in RCNN has the same number of feature maps as the corresponding GRCL. Batch normalization is also inserted after each convolutional kernel in RCL. We fix $T$, and compare these two models. The results in Table 2(a) show that each GRCNN model outperforms the corresponding RCNN on both IIIT5K and SVT. Those results show the advantage of the introduced gate in the model.

Furthermore, we explore the effect of iterations in GRCL. From Table 2(a), we can conclude that having more iterations is beneficial to GRCL. The increments of accuracy between each iteration number are 1.50%, 0.79% on IIIT5K and 1.22%, 1.09% on SVT, respectively. This is reasonable since GRCL with more iterations is able to receive more context information.

Finally, we compare various peephole LSTM units in the bidirectional LSTM for processing feature sequences. Five types of LSTM variants are compared: full peephole LSTM (LSTM-A), input gate peephole LSTM (LSTM-I), output gate peephole LSTM (LSTM-O), forget gate peephole LSTM (LSTM-F) and none peephole LSTM (LSTM). In the feature extraction part, we use GRCNN with $T = 1$ which is described in Table 1. Table 2(b) shows that the LSTM without peephole connections ($\gamma_1 = \gamma_2 = \gamma_3 = 0$) gives the best result.

## 4.4 Comparison with the state-of-the-art

We use the GRCNN described in Table 1 as the feature extractor. Since having more iterations is helpful to GRCL, we use GRCL with T = 5 in GRCNN. To fairly compare the GRCNN with other network architectures, such as the ResNet or the untied RCNN [21], we also untie the weights in the recurrent part as [21] did. For sequence learning, we use the bidirectional LSTM without peephole connections. The training details are described in Sec.4.2. The best model on the validation set of Synth90k is selected for comparison. Note that this model is trained on the training set of Synth90k and not finetuned with respect to any dataset. Table 3 shows the results. The proposed method outperforms most existing models for both constrained and unconstrained text recognition.

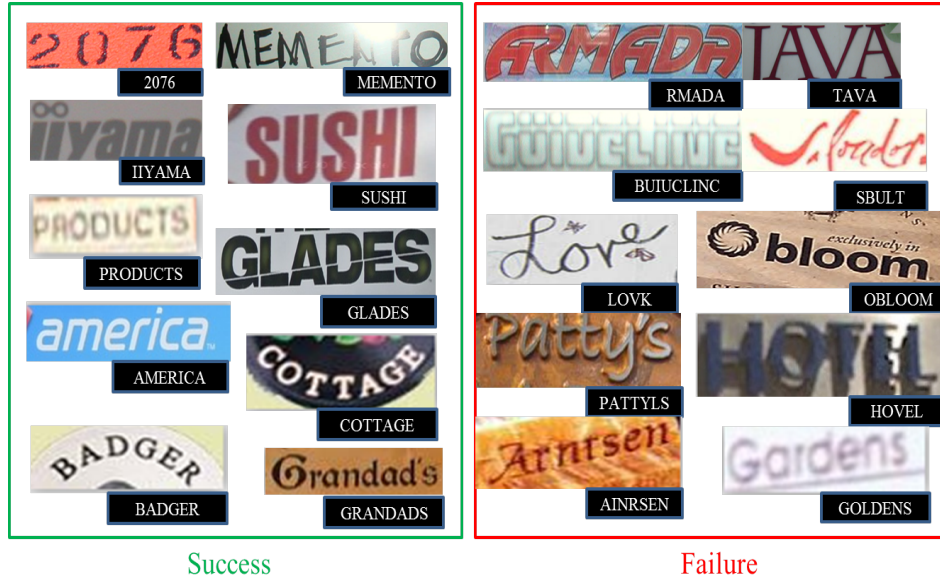

<div align="center">

Success                                              Failure

</div>

Figure 4: Lexicon-free recognition results by the proposed GRCL-BLSTM framework on SVT, ICDAR03 and IIIT5K

Moreover, we do an extra experiment by building a residual block in feature extraction part. We use the ResNet-20 [9] in this experiment, since it has similar depth with GRCNN with $T = 5$. The implementation is similar to what we have discussed in Sec.4.2. The results of ResNet-BLSTM are listed in Table 3. This ResNet-based framework performs worse than the GRCNN-based framework. The result indicates that GRCNN is more suitable for scene text recognition.

Some examples predicted by the proposed method under unconstrained scenario are shown in Figure 4. The correctly predicted examples are shown in the left. It is seen that our model can recognize some long words that have missing part, such as "MEMENTO" in which "N" is not clear. Some other bend words are also recognized perfectly, for instance, "COTTAGE" and "BADGER". However, there are some words that cannot be distinguished precisely and some of them are showed in the right side in Figure 4. The characters which are combined closely may lead to bad recognition results. For the word "ARMADA", the network cannot accurately split "A" and "R", leading to a missing character in the result. Moreover, some special symbols whose shapes are similar to the English characters affect the results. For example, the symbol in "BLOOM" looks like the character "O", and the word is incorrectly recognized as "OBLOOM". Finally, some words that have strange-shaped character are also difficult to be recognized, such as "SBULT" (the ground truth is "SALVADOR").

## 5   Conclusion

we propose a new architecture named GRCNN which uses a gate to modulate recurrent connections in a previous model RCNN. GRCNN is able to choose the context information dynamically and combine the feed-forward part with recurrent part flexibly. The unrelated context information coming from the recurrent part is inhibited by the gate. In addition, through experiments we find the LSTM without peephole connection is suitable for scene text recognition. The experiments on scene text recognition benchmarks demonstrate the superior performance of the proposed method.

## Acknowledgements

This work was supported in part by the National Basic Research Program (973 Program) of China under grant no. 2013CB329403, the National Natural Science Foundation of China under grant nos. 91420201, 61332007, 61621136008 and 61620106010, and in part by a grant from Sensetime.

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
