[Reviews · NeurIPS 2017]

Reviewer 1



This paper introduces a modification to an existing network architecture, adding in gating to recurrent convolutional networks and applies this to scene text recognition. The method shows modest improvements over previous state of the art results. This is a difficult problem and to achieve across the board improvements on all datasets is a very hard task. The architecture adds some small novelties and trained cleanly end-to-end, with nice ablation studies in Table 2 so I think this is an important method for this area, and these ideas could transfer to other image analysis tasks. I am going to raise my rating to a 7.

Reviewer 2



This paper introduces gates to the RCNN layer and shows positive improvements over vanilla RCNN architectures on a wide range of pre-detected text line recognition benchmarks. This paper doesn’t try to address the end-to-end problem of text extraction from natural images, where detection needs to be combined with recognition. The authors did a very rigorous analysis on different challenging benchmarks compared to their baseline and show SOTA results in many of them. But the paper has too many grammatical and orthographic errors for a NIPS paper. I encourage the authors to give a more thorough review. Also, the idea proposed by the paper which is to introduce a gating mechanism for recurrent output of the layer makes sense but is not sufficiently novel on its own. The combination of this and just decent improvement over previous SOTA makes it just a small positive contribution. The author could potentially modify the scope of the paper to include the detection piece of the problem of extracting text from natural images. If they show more improvements on this end-to-end system, it would be more interesting as the community has not yet figured out very good systems on such end-to-end tasks [1]. Another approach is to show that this gating idea is more general, meaning that it can be used for non-OCR problems. I would encourage to try this on the ImageNet benchmark and potentially another problem (detection, depth estimation, etc.). [1] Uber-Text: A Large-Scale Dataset for Optical Character Recognition from Street-Level Imagery

Reviewer 3



Authors added gating mechanism in Recurrent Convolutional NN that allow controlling the amount of context / recursion. In other words, it is similar to improvement from RNN to GRU but for Recurrent CNNs. Authors benchmark their model on text reading task. Positives aspects: - authors improved upon the state of the art results on very well research benchmark - the proposed model combines few proved ideas into a single model and made it work. Areas to improve: - Lines 195-196: you state that choose 10 best models per architecture that perform well on each dataset and state the results as the median of them. It is ok when you compare different iterations of you model to make sure you found good hyperparameters. But this cannot be at the final table with the comparison to other works. You should clearly state what is your pool of models (is it coming from different hyperparameters? what parameters and what values did you choose). How many models did you try during this search? You should provide single hyper parameter + model setup with the results that works the best across all the problems in the final table. Additionally, you may provide variance of the results with the clear explanation of hyper parameter search you performed. - did authors try to use attention not bidirectional RNN + CTC loss? In few recent works (https://arxiv.org/pdf/1603.03101.pdf, https://arxiv.org/pdf/1704.03549.pdf, https://arxiv.org/pdf/1603.03915.pdf). This should still allow for improving the results, or if didn't work, this is worth stating. I vote for top 50% of papers as improving upon the previous state of the art results on such a benchmark is a difficult job. Authors proposed a new model for extracting image features that is potentially beneficial to other visual tasks. This is conditional on fixing the results number (to present numbers from a single model, not median) and open-sourcing the code and models to the community for reproducibility.